# The Effectiveness of Clove Extract on Oxidization-Induced Changes of Structure and Gelation in Porcine Myofibrillar Protein

**DOI:** 10.3390/foods11131970

**Published:** 2022-07-02

**Authors:** Jinming Ma, Deyin Pan, Ying Dong, Jingjing Diao, Hongsheng Chen

**Affiliations:** 1College of Food Science, Heilongjiang Bayi Agricultural University, Daqing 163319, China; jinmingma@byau.edu.cn (J.M.); pandeyin@byau.edu.cn (D.P.); 2Huangpu Customs Technology Center, Dongguan 523000, China; dongying51@126.com; 3National Coarse Cereals Engineering Research Center, Heilongjiang Bayi Agricultural University, Daqing 163319, China; diaojing62@163.com; 4China-Canada Cooperation Agri-Food Research Center of Heilongjiang Province, Daqing 163319, China

**Keywords:** myofibrillar proteins, clove extract, antioxidants, protein structure, gelation

## Abstract

This study aimed to investigate the structural characteristics and gelation behavior of myofibrillar proteins (MPs) with or without clove extract (CE) at different oxidation times (0, 1, 3, and 5 h). Circular dichroism spectra and Fourier transform infrared spectra showed that samples with CE addition had significantly higher α-helix content after oxidation than those without CE addition. However, prolonged oxidation (5 h) would make the effect of CE addition less pronounced. Similarly, the ultraviolet-visible (UV) spectra analysis revealed that CE controlled the oxidative stretching of the protein tertiary structure and reduced the exposure of aromatic amino acids. In addition, the particle size and turbidity values of the CE group significantly decreased after oxidation compared to the non-CE group. CE increased the gel strength by 10.05% after 5 h of oxidation, which could be observed by scanning electron microscopy (SEM) as a more homogeneous, dense, less porous, network-like gel structure. Therefore, these results showed that oxidation induced significant changes in the structure and gel properties of MPs, but the addition of CE effectively inhibited these destructive changes.

## 1. Introduction

Myofibrillar proteins (MPs) are the most abundant source of proteins, including myosin, actin, tropomyosin, and troponin, accounting for approximately 55–60% of muscle protein and play a crucial role in the structural properties of muscle. Among them, myosin and actin contribute the most to the gelling properties [1], which favors the formation of gels with palatability or good sensory perception. However, proteins are highly susceptible to oxidation during processing and storage, leading to a decrease in their functional properties and nutritional quality [2]. Therefore, it is vital to find a method to reduce protein oxidation. Adding an appropriate amount of polyphenol-containing spice extracts and natural antioxidants with the advantages of “safety and non-toxicity” is a method to retard protein oxidation, which has become a current focus.

There are three common protein oxidation systems in meat and meat products: hydroxyl radical-generating system (HRGS), fat oxidation product-induced oxidation system and high iron myoglobin oxidation system, among which the hydroxyl radical oxidation system is one of the most prevalent oxidation systems in meat products processing, and hydroxyl radical is also the most powerful oxidizing radical among reactive oxygen radicals [3]. Many scholars used the hydroxyl radical system to simulate the oxidation of meat products. Park et al. [4] reported the concentration effects of HRGS on biochemical properties of porcine muscle myofibrillar protein, and our previous study [5] found that the effectiveness of clove extracts in the inhibition of HRGS induced structural and rheological changes in porcine myofibrillar protein. HRGS oxidation can lead to meat quality deterioration, such as reduced tenderness and juiciness, decreased flavor and color acceptability, and the formation of toxic compounds [5,6,7]. HRGS oxidation also promotes the production of carbonylation and intermolecular cross-linking, all of which can influence protein’s functional properties [3]. Researchers have adopted various methods to reduce protein oxidation, including dietary strategies, low temperatures, and food additives [8,9]. Among these, antioxidants act as food additives, effectively reducing protein oxidation and hindering the deterioration of the quality of meat products [10]. Natural antioxidants extracted from plant sources have become popular, as they present excellent properties such as free radical-scavenging ability, metal ion chelation activity, and singlet oxygen-quenching ability [11,12]. Spices and herbs are known to exhibit practical activities to inhibit lipid and protein oxidation in various food systems [13].

Spice extracts (clove, licorice, rosemary, nutmeg, ground cardamom, and cassia bark, etc.) have a high total phenolic content, which is a potent inhibitor in the production of thiobarbituric acid reactive substances in cooked pork patties [5]. Notably, clove extract (CE) isolated from clove (*Syzygium aromaticum*) buds has antioxidant, antibacterial, antiviral, and anti-inflammatory properties, as well as other physiological functions, and it has been widely used in medicine, healthcare, food flavoring, and cosmetics [14]. Polyphenols, whose oxygen atoms on adjacent hydroxyl or carbonyl groups can cooperate with metal ions as paired atoms to form five- or six-membered chelates. Two adjacent phenolic hydroxyl groups can form stable five-membered chelates with metal ions in the form of oxygen negative ions. When plant polyphenols interact with some high-valent metal ions, they can reduce the metal ions from high-valent to low-valent states while complexing them. The complexation of plant polyphenols with metal ions, such as Cu^2+^ and Zn^2+^, results in the inhibition of metalloenzymatic activities containing these ions [15]. Our previous study found that in CE, the main antioxidant ingredient is eugenol (58.40%). It accounted for the largest percentage of CE. Eugenol has been found to be equivalent to vitamin E as a natural antioxidant [16]. Eugenol has been reported to scavenge free radicals by transferring electrons and terminating the chain reaction of free radicals [17]. Recently, the addition of CE in meat processing has been partially studied to improve the quality of meat products [18,19,20]. For example, Krishnan group [21] reported that CE could effectively inhibit lipid oxidation and microorganism growth, enhancing the sensory quality and shelf life of refrigerated raw chicken meat. Shan et al. [22] proposed that CE could help reduce total bacterial counts and inhibit the lipid oxidation of raw pork at room temperature for nine days, revealing that CE exhibited the highest antioxidant activity and antibacterial properties. A study by Zhang et al. [23] found that CE inhibited the increase in lipid oxidation product content and protein carbonyl formation in Chinese-style sausages by inhibiting lipid peroxidation and scavenging free radicals. Armenteros et al. [24] found that CE inhibited protein oxidation in cooked ham during refrigeration. Our previous study also indicated that CE displayed significant reducing power and vigorous DPPH scavenging ability [10].

This study builds on our previous findings that of the four spices, namely CE, rosemary extract, cinnamon extract, and licorice extract [25], CE is the most effective in suppressing protein oxidation and has a highly concentration-dependent nature. Moreover, our previous results also showed that compared with the control at the oxidation 5 h, the addition of CE significantly decreased the carbonyl formation by 44.07%, enhanced the solubility by 20.82%, and improved the gel formation ability (storage modulus, loss modulus) and thermal stability of MPs. The protective effect of CE on protein denaturation was reflected in keeping Ca-ATPase activity and decreasing protein aggregation. The result of SDS-PAGE showed that with the increasing oxidation time, the band intensity of control samples decreased more rapidly than that of the CE samples, indicating that CE can scavenge hydroxyl radicals, so the polymerization and aggregation can be controlled by CE to some extent [5]. To further investigate the effect of CE on the structure and gelation of oxidized MPs, ultraviolet-visible (UV), circular dichroism (CD), and Fourier transform infrared (FTIR) spectra were used to illustrate the conformational changes of MPs under HRGS oxidation. The aggregation of proteins under oxidation or non-oxidation and with or without CE was analyzed by particle size and turbidity. Finally, the gel-forming capacity of MPs was studied by gel strength and scanning electron microscopy (SEM). This provides a concrete theoretical basis and a novel approach to rationally apply CE to muscle foods.

## 2. Materials and Methods

### 2.1. Materials

Ten entire carcasses were randomly selected from a large population based on criteria for age (5–6 months) in Beidahuang Meat Processing Co., Ltd. (Harbin, China). Ninety pieces of individual pork longissimus lumborum muscle (purchased in different days, approx. 150 g) were vacuum-packaged from each animal. Pork longissimus muscle (core temperature of 4 °C, pH 5.8–6.2) was purchased less than 24 h post-mortem from Beidahuang Meat Processing Co., Ltd. and kept on ice during transport to the meat science laboratory at Heilongjiang Bayi Agricultural University. Clove buds were purchased from the local pharmacy (Harbin, China). The dried clove buds were ground to a fine powder using the Kenwood Multi-Mill (Kenwood Ltd., Havant, UK) and passed through a 24-mesh (700 μm) sieve. Bromatum Kalium, glutaraldehyde, phosphate, chloroform, ethylenediaminetetraacetic acid (EDTA), Trolox C, ascorbic acid, propyl gallate, and piperazine-N,N′-bis (2-hydroxypropanesulfonic acid) (PIPES) were purchased from Sigma Chemical Co., Ltd. (St. Louis, MO, USA). All reagents and chemicals were of analytical grade.

### 2.2. Preparation of CE

CE was prepared as reported in our previous work [26]. A total of 100 g of dried and crushed cloves was mixed with 800 mL of 95% (*v*/*v*) ethanol in a triangular flask. After being placed in a constant temperature incubation shaker (100 rpm) at 55 °C for 12 h, the extract was filtered using filter paper, and the residue was filtered by re-extraction with 400 mL of 95% ethanol for 12 h. The combined liquids were freeze-dried after concentration in a vacuum through a rotary evaporator (50 °C). The extracts were stored in a refrigerator at 4 °C.

### 2.3. Preparation of MPs

The MPs were extracted according to the method of Xia et al. [27], stored at 4 °C, and used within 36 h. The biuret method was used to detect the protein concentration of the MPs. The resulting MPs were stored at 4 °C and used within 24 h.

### 2.4. Oxidation of MPs Samples and Experimental Design

MP concentration was adjusted to 20 mg/mL with 15 mmol/L PIPES buffer and 0.6 mol/L NaCl (pH 6.25), and the concentration of CE was 1.0 mg/mL (CE dry weight/solution volume). The MP samples with or without CE were oxidized by HRGS buffer composed of 10 μmol/L FeCl_3_, 10 mmol/L H_2_O_2_, and 0.1 mmol/L ascorbic acid (4 °C; 0, 1, 3, or 5 h). Oxidation was terminated by the addition of the mixture of EDTA, Trolox C, and propyl gallate (1 mmol/L each). The non-oxidized MP solution contained EDTA, Trolox C, and propyl gallate as a control group.

### 2.5. UV Scanning Spectra

The conformational changes of MP were measured at room temperature with a UV-spectrophotometer (UV-1800, Shimadzu, Japan) according to a slightly modified method by Wang et al. [28]. MPs solutions (1 mg/mL) were scanned over a range of 200–400 nm at 1 nm increments with PIPES buffer (15 mmol/L, pH 6.25 in 0.6 mol/L NaCl) as the blank control, and the curves obtained were smoothed three times.

### 2.6. Secondary Structure of MPs

CD spectrum analysis was performed to analyze the secondary structure of the MP samples according to the method of Chen et al. [29]. Oxidized or non-oxidized MP samples were diluted to 50 μg/mL with 15 mmol/L PIPES (pH 6.25) and 0.6 mol/L NaCl. Molecular ellipticity was measured in the range of 190–250 nm at 25 °C, with deionized water as control, the scan rate was 100 nm/min, and the interval time was 0.25 s.

### 2.7. Fourier Transform Infrared (FTIR) Spectra

FTIR spectra could reflect the secondary structure of proteins [30]. The MP samples (dry form) were measured by FTIR spectra with some modifications based on the reported method by Sun et al. [31]. The samples were mixed with bromatum Kalium, and the mixture was ground and pressed into a disc. FTIR spectra were obtained in the wavenumber range from 400 to 4000 cm^−1^. Detection was carried out at 20 °C, and the infrared image of each sample was a superimposed image of multiple scans. The results were analyzed by EZ-Ominic software and Peak Fit Version 4.12 software (SPSS Inc., Chicago, IL, USA).

### 2.8. Particle Size

The protein concentration of all the MP samples was diluted to 5 mg/mL with 15 mmol/L PIPES (pH 6.25) containing 0.6 mol/L NaCl. The particle size distribution was held using the Malvern Master sizer (Malvern Instruments Co. Ltd., Worcestershire, UK) reported by Zhong et al. [32].

### 2.9. Protein Turbidity

Turbidity measurement was performed according to Pan et al. [33] with slight modifications. A 5 mL aliquot of the protein samples (1 mg/mL) was measured in water baths at 30, 40, 50, 60, 70, and 80 °C for 30 min. The absorbance at 600 nm was obtained using a spectrophotometer (model 200PLUS, Analytik Jena AG, Jena, Germany). Turbidity was expressed as the absorbance value.

### 2.10. Determination of Protein Gel Strength

Gel strength measurement was performed according to Jia et al. [34]. Glass vials (inner diameter × length = 25 mm × 40 mm) contained 7 g of MPs solution (40 mg/mL in 15 mmol/L PIPES buffer, containing 0.6 mol/L NaCl, pH 6.25). Each treatment was repeated three times, covered lightly with matching threaded plastic caps, and placed in a 70 °C water bath for 20 min. Then, the samples were cooled in an ice–water mixture for 1 h and then placed in a 4 °C freezer overnight. After 30 min of equilibrating at room temperature (20–25 °C), the gel strength was measured using P/0.5 in the TA.XT plusC Texture Analyzer (Stable Micro Systems Ltd., Surrey, UK), and the speed before the test was set to 2.0 mm/s; the test speed was 0.3 mm/s; the speed after the test was 2.0 mm/s; and the puncture distance was 10.0 mm.

### 2.11. Scanning Electron Microscopy

Pre-treatments of the MPs gel samples were performed according to the method of Xia et al. [35] with minor modifications. Subsequently, the samples were analyzed using an S-3400N Scanning Electron Microscope (Hitachi High Technologies Corp., Tokyo, Japan). The SEM images were analyzed at a magnification of 200×.

### 2.12. Statistical Analysis

Results are presented as mean ± standard error (SE) and statistically analyzed using the General Linear Models procedure of the Statistix 9.0 software package (Analytical Software, St. Paul, MN, USA). Analysis of variance (ANOVA) with Tukey’s multiple comparisons was used to measure the significance of the main effects (*p* < 0.05). Three batches of MPs samples (replicates) were produced to perform UV, CD, FTIR spectra, particle size changes, protein turbidity, protein gel strength, and electron microscopy analysis. All experiments were conducted in triplicate (triplicate observations) for each batch of the MPs samples. Data were analyzed using the mixed procedure, in which replicates (*n* = 3) were included as random effects; moreover, the different treatments and oxidation time (0, 1, 3 and 5 h) were included as fixed terms.

## 3. Results and Discussion

### 3.1. Changes in UV Spectra

In the near-ultraviolet region (200–400 nm), Trp, Tyr, and Phe have different UV absorption spectra due to their different chromogenic groups (indole group for Trp, phenol group for Tyr, and phenyl group for Phe). Thus, UV spectra can be used to determine proteins and study structural changes in amino acids [28].

As shown in Figure 1A, the shape and intensity of the peaks in each spectrum changed significantly with increasing oxidation time, indicating that oxidation significantly destroyed the structures of amino acids and caused structural variations in amino acids containing benzene rings or conformational modifications in proteins [36,37], resulting in a significant decrease in their UV absorption in solution. In the 270–280 nm range, the intensity of the peaks of the oxidized protein samples was markedly weakened, and their positions were altered considerably, which was possibly due to a decrease in tyrosine or tryptophan content. Compared with the control group samples (non-oxidized) (λ_max_ = 276 nm, OD = 1.65), after oxidation for 5 h, both the λ_max_ of the CE (λ_max_ = 278 nm, OD = 1.03) and non-CE group (λ_max_ = 278 nm, OD = 0.82) are redshifted to 278 nm at the absorption peak; the resulting redshift could be due to the significant reduction in light absorption intensity after oxidation [27]. Our previous findings indicate that the oxidation of tyrosine forms dimerized tyrosine, which reduces the content of tyrosine monomers. Moreover, the NH group in the indole structure of tryptophan is also easily oxidized, resulting in a decrease in its content [25]. Therefore, this fully demonstrates that oxidation changes the structure of tyrosine or tryptophan and explains the decreased peak intensity at 270–280 nm in the UV spectrum. Wang et al. showed that a higher degree of protein oxidation occurred during microwave thawing of pork loin, which promoted protein unfolding and also caused a redshift in λ_max_ [38]. However, the addition of CE significantly suppressed the decrease in light absorption intensity. CE-treated samples (oxidation 1, 3 and 5 h later) showed an increase in UV absorption peaks compared to that of the non-CE samples. The increased UV absorption peaks are attributed to the fact that CE can protect the structure of aromatic amino acids from oxidative damage. In addition, each sample also produced strong light absorption in the 200 to 220 nm range. UV absorption in this range may be caused by some small peptide molecules [39]. In this study, no significant change in peak shape was detected in the peak intensities of each group of samples within this range.

As it is difficult to distinguish the specific characteristics of the peaks due to the superposition of the spectral peak signals, at present, derivatization of the ultraviolet absorption spectrum to obtain the second derivative spectrum is the most effective method to distinguish the specific characteristics of the peak. The ratio (r = a/b), which represented the peak-to-valley values in the second derivative spectrum [40,41], was also observed to reflect oxidation destroyed the structures of tyrosine and tryptophan in Figure 1B. The “r” values of the oxidation-treated samples for 1, 3, and 5 h (1.62, 1.71, and 1.71) decreased by 16.1, 19.3, and 18.1%, compared with the control group samples (1.36), respectively. An increase in the value of “r” is thought to indicate that the tertiary structure of the protein has unfolded to such an extent that more tyrosine or tryptophan residues are exposed [42]. The addition of CE significantly controlled the oxidative elongation of the protein tertiary structure and reduced the exposure of aromatic amino acids. Recently, fluorescence spectroscopy is a good method to determine the tertiary structural changes of proteins; the next study will be conducted by Trp fluorescence to investigate the effect of CE on the tertiary structure of oxidized MPs in the future.

### 3.2. Changes in CD Spectra

The changes in the secondary structure of MPs’ samples with different treatments were detected by CD spectra. Figure 2A shows that there are two negative peaks around 210 and 220 nm, implying the predominant presence of α-helix conformation due to the dimer of myosin tails with two intertwined α-helices [43,44]. In this band region, the distinct spiral pattern of the non-oxidized samples was the lowest. In contrast, the peak intensity of the oxidized MPs increased significantly (*p* < 0.05), especially in the case of the 3 and 5 h oxidized MPs samples without CE. The peak intensity of the 3 and 5 h oxidized MPs samples with CE was significantly lower than that of the samples without CE at the same time. These results suggest that the losses in the α-helix pattern were caused by oxidation, and CE plays a crucial role in protecting the α-helix pattern of MPs from oxidative damage to some extent. Sun et al. [45] indicated that the percentage of α-helix conformation of the protein decreased after oxidation, which agrees with our finding. Moreover, the effect of CE on the α-helix content of oxidized MPs is shown in Figure 2B. The α-helix content of the MPs’ samples without CE decreased significantly after oxidation; the denaturation of protein molecules will affect the decline of α-helix [46], while that of oxidized MPs (3 and 5 h with CE) was significantly higher than that without CE. Shen et al. [47] believed that the increase in α-helix content may be caused by the binding of phenolic compounds driven by electrostatic and hydrophobic interactions with MPs, thereby improving the stability of MPs. These data demonstrated that the incorporation of CE favors the protection of MPs from oxidation. However, there was no significant difference between the samples oxidized 1 h with CE and without CE. This is attributed to the fact that there was insufficient time for the antioxidant compounds of CE to reach the interior of the protein to work. A similar phenomenon has been observed by many other studies on the different effects of different concentrations of phenolics on different proteins [43,48,49].

### 3.3. Changes in FTIR Spectra

FTIR spectra analysis was adopted to determine the secondary structure of the myofibrillar protein samples. Generally, the secondary structure of the protein is maintained by the hydrogen bond networks of C=O and N–H on the main chain, which contributes to the changes in the infrared amide I spectrum (1600–1700 cm^−1^). Thus, we focused on the absorption range of 1600–1700 cm^−1^ in the FTIR spectra, as depicted in Figure 3. We can see the stacked deconvoluted curve of the amide I spectrum of the MPs and the eight major bands associated with protein conformation. The band located at 1655.3 cm^−1^ is assigned to the α-helix. Visibly, this band for the oxidized MPs samples shifted to the left to a different extent. Specifically, those for the MPs samples oxidized for 1, 3, or 5 h without CE shifted to 1656.9, 1656.5, and 1658.7 cm^−1^, and those oxidized for 1, 3, or 5 h with CE shifted to 1655.6, 1655.7, and 1659.4 cm^−1^, respectively. The results demonstrated that the electron density of carbonyl oxygen increased and the hydrogen bond strength decreased, which is in agreement with previous studies [5,31].

The influence of CE on the secondary structure content of oxidized MPs samples is listed in Table 1. The degree of α-helix content variation for different treatments was different. Specifically, the α-helix of the MPs gradually decreased, with the oxidation time increasing. There is a significant reduction in the α-helix content of the MPs without CE. Whereas, in the case of CE treatments, the degree of α-helix content reduction is little during the first 3 h, but there is a significant decrease in the next 2 h. These results indicated that the addition of CE inhibited the reduction in α-helix; however, when the oxidation time was prolonged, up to 5 h, CE could not work, which is consistent with the results of CD spectra.

In Figure 3, the bands located at 1635.2, 1675.6, and 1686.6 cm^−1^ originate from the vibration of the anti-parallel β-sheet structure; the band at 1625.4 cm^−1^ originates from the aggregated intermolecular β-sheet; that at 1665.2 cm^−1^ corresponds to the β-turn structure; the one at 1645.1 cm^−1^ was generated from the random coil structure, and the one at 1614.7 cm^−1^ may be assigned to the vibration of the aromatic ring of tyrosine residues. In this study, the β-sheet content decreased after oxidation, while that with CE remained approximately the same as that of the control group. Additionally, both of the secondary structures of the MPs with and without CE changed (*p* < 0.05) after 5 h oxidation. Hence, it can be concluded that the addition of CE might change the oxidation process of HRGS. As shown in Figure 3, the position of the infrared absorption peaks of the 1 h oxidized MPs samples without CE changed remarkably compared to the non-oxidized MPs, which is attributed to the structure of MPs, which is sensitive to foreign HRGS. Consequently, the secondary structure of MPs changed noticeably within a short time with oxidative stress, leading to the α-helix content decreasing and the β-sheet and random coil structure increasing. As the degree of oxidation increased, the secondary structure content became balanced. These results demonstrated that the primary changes in the secondary structure of MPs occur within seconds, which are mainly caused by oxidative stress. Luckily, the addition of CE inhibited the structural changes, which is attributed to the CE partially offsetting the influence from oxidation or the CE-occupied interaction sites of HRGS with MPs.

### 3.4. Particle Size

Figure 4 presents the particle sizes of all the samples sorted from 10 to 1000 μm. As shown in Figure 4A, there are some protein molecules whose particle size is less than 10 μm in the non-oxidized MPs samples, while all these small proteins aggregated to form a polymer under oxidation. Notably, in Figure 4B, some protein molecules still had a particle size below 10 μm in the CE treatments. After the MPs were oxidized, rightward shifts occurred in all peaks, resulting in the particle size moving to the increasing direction. Specifically, the largest shift occurred in the 5 h oxidized samples without CE. In the case of the CE-containing oxidized MPs, the extent of the left shift of the three peaks was approximately the same. However, when comparing the 5 h oxidized MPs sample with CE with the sample without CE, the particle volume percentage of the former was visibly lower than that of the latter. This result demonstrated that the introduction of CE inhibited the formation of macromolecular protein aggregates. Moreover, aggregates larger than 1000 μm were found in the samples oxidized 5 h without CE, which may have originated mainly from disulfide bonds or hydrophobic interactions [50]. However, no aggregate whose particle size was larger than 1000 μm was formed in the samples oxidized for 5 h with CE, demonstrating that the addition of CE prohibited the oxidative polymerization of MPs.

Generally, the thermal denaturation of the protein would lead to protein aggregation and polymer formation. The MPs tended to aggregate after oxidation, whereas the addition of CE inhibited the formation of myofibrillar protein polymers, which is consistent with the particle size distribution. Schmitt et al. [51] found that the particle size of polymers can change the functional properties of the protein, such that the gel texture with large particle size polymers was worse than that with small ones. Subsequently, large particle size polymers would reduce the water-holding capacity of the gel. The above results suggest that CE protected the MPs from HRGS oxidation, namely, the weak oxidative stress that occurred in the MPs was attributed to the introduction of CE.

### 3.5. MPs Turbidity

Protein turbidity is a vital index that can reflect the change in the thermal stability of protein after oxidation. The higher the turbidity value is, the higher the degree of protein aggregation [52]. As shown in Figure 5, the turbidity of all samples increased significantly with increasing temperature (*p* < 0.05); simultaneously, as the oxidation time increased, the turbidity value also increased significantly (*p* < 0.05). This result could be attributed to the fact that oxidation reduces the thermal stability of proteins and alters the protein dissolution equilibrium, resulting in protein denaturation and aggregation. Therefore, the increase in temperature and oxidation time can increase the degree of protein chaos. Our previous study found that the T_max1_, T_max2_, and T_max3_ of fresh muscle MPs were 60.7, 69.0, and 78.8 °C, respectively, after oxidation for 1 h and 3 h, the T_max2_ and T_max3_ in the CE group were significantly lower than those in the fresh muscle MPs group (*p* < 0.05). The T_max1_ disappeared in both the control group and CE group after oxidation, suggesting that the head of myosin is more easily oxidized than the tail and actin. So, the oxidation reduces the thermal stability of MPs and CE could protect the MPs’ structure [5]. However, the turbidity value of the CE-treated group was significantly lower than that of the non-CE group at the same oxidation time (*p* < 0.05). Typically, at 80 °C, compared with the non-CE group, the turbidity value of the CE-treated group after 1, 3, and 5 h decreased by 8.67, 15.47, and 14.44%, respectively. In that case, the CE-treated MPs’ solution contains many smaller particles. The lower turbidity value is derived from the reduction in light scattering in the solution [47]. This result indicates that the addition of CE can effectively enhance the thermal stability of the protein and control protein denaturation and aggregation, and this is consistent with the surface hydrophobicity results of our previous studies [25].

### 3.6. Gel Strength

Gel strength is one of the crucial indicators for evaluating muscle protein gel properties, and MPs have the most significant heat-induced gelling ability among animal proteins [53]. As shown in Figure 6, the gel strength of each sample decreased gradually with increasing oxidation time. Compared with the non-oxidized samples, the gel strength of the oxidized samples for 1 h showed no significant difference (*p* > 0.05). However, the gel strength of the oxidized samples decreased significantly (*p* < 0.05) at 3 h and 5 h. This result indicates that short-time oxidation has little negative effect on gel strength, but long-time oxidation greatly hinders the formation of the gel network structure. When oxidized for 3 h and 5 h, the gel strength of the CE group was significantly higher than that of the non-CE group (*p* < 0.05). Compared with the non-CE group, the gel strength increased by 9.79% for the 3 h samples and 10.05% for the 5 h samples. This result implies that the addition of CE can significantly inhibit oxidative damage of proteins and improve gel strength.

During the heating of MPs to form a gel, the myosin head aggregates first, and then the tail continues to aggregate to form a network structure [3], while after oxidation, the aggregation pattern of myosin is dominated by tail–tail cross-linking [54]; therefore, it is likely that when oxidized, MP is heated to form a different quality gel. Numerous studies have shown that mild protein oxidation can improve the protein gel network by changing the arrangement of myosin, while excessive oxidation can produce a large number of aggregates that cannot form a dense gel network structure [55,56,57]. Lund et al. [58] showed that the oxidation of thiol groups of proteins transformed into intermolecular disulfide bonds and affected the stiffness and elasticity of protein gels. The addition of CE promotes protein unfolding, as confirmed by the exposure of hydrophobic groups in previous studies [25], which may disrupt the hydrogen bonding of α-helix and promote the conversion of α-helix to other three secondary structures. The appropriate extent of protein unfolding and conversion of α-helix to β-sheet were beneficial for gelation. The results showed that the clove extract protected the reduction in β-sheet structure and improved the gel strength.

Li et al. [59] found that the gel strength of MPs can be enhanced by hydrogen bonding and hydrophobic interactions between MPs and the phenols from spice extracts. Other studies have found that oxidized phenolic compounds formed during gel preparation can generate covalent interactions and enhance cross-linking [43]. Water molecules in the gel can be locked by non-covalent interactions such as hydrogen bonds produced in phenolic extracts [60], which is more favorable for improving gel strength.

### 3.7. Scanning Electron Microscopy

Figure 7 shows the effect on the microstructure of myofibrillar protein gels under different oxidation times (1, 3, and 5 h) with or without CE. The fresh muscle protein gels exhibit a homogeneous, sturdy network structure in Figure 7A. The gaps in the gels appeared and gradually increased and became larger with increasing oxidation time, and the smooth gel structure became rougher and rougher compared to the initial gel of the fresh MPs. Especially at 3 h and 5 h for the non-CE samples, oxidation caused the gel network to collapse, resulting in large, irregular cracks in the gel (Figure 7C,D). These changes in microstructure may explain the deterioration of gel strength after a long oxidation time. After 3 and 5 h of oxidation, compared to the non-CE group, the gel with CE showed a more uniform and dense structure, with smaller pores and a more continuous and intact network matrix. In addition, such a dense gel structure could withstand the puncture force, thus improving the gel strength in the same oxidation time (Figure 6). These results indicated that CE could inhibit the inferior gel formation caused by protein oxidation (Figure 7b–d). While the gel structure is protected to a certain extent by the addition of CE, the gel becomes relatively smooth and dense, and there are more covalent and non-covalent connections within this gel, which caused the gel to be able to withstand larger destructive forces and the gel strength to be higher. In addition, the rheological results [5] were consistent with the above results; therefore, these results indicated that CE could protect the gel network structure from oxidative damage.

## 4. Conclusions

In this work, the structural and gelling property changes of a series of oxidized MPs samples with/without CE induced by hydroxyl radicals were investigated. The results demonstrated that the addition of CE can significantly inhibit the decrease in UV absorption intensity and suppress the reductions of α-helix, β-sheet and β-turn content caused by oxidation, indicating that CE has a protective effect against the oxidative damage of the aromatic amino acid structure in the samples. Moreover, the addition of CE significantly inhibited the formation of large protein aggregates, improved the gel-forming ability, and inhibited the oxidative deterioration of the gel microstructure. This work provides us with an in-depth understanding of the influence of CE on the structural and functional properties of MPs, which could benefit the design of potent antioxidants to protect essential muscle food proteins.

## Figures and Tables

**Figure 1 foods-11-01970-f001:**
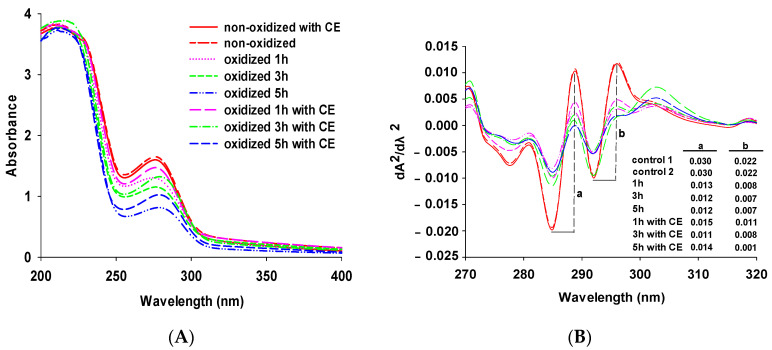
Effect of clove extract (CE) on the ultraviolet-visible (UV) spectrum of MPs under (**A**) different oxidation times and the spectrum of the second derivative of the UV scan (**B**). Letters “a” and “b” indicate the peak-to-valley values of the two main peaks. Control 1 means non-oxidized and Control 2 means non-oxidized with CE.

**Figure 2 foods-11-01970-f002:**
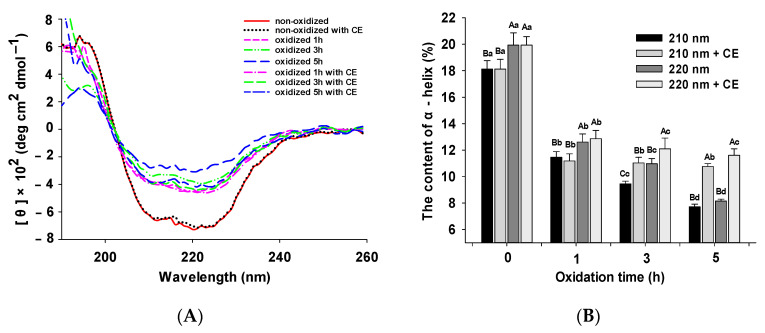
Effect of CE on circular dichroism (CD) spectrum (**A**) and α-helix content (**B**) of MPs under different oxidation times; [θ] is the mean residue ellipticity. Different capital letters indicate significant differences between different treatments under the same oxidation time (*p* < 0.05); different lowercase letters indicate the same treatment with significant differences at different oxidation times (*p* < 0.05).

**Figure 3 foods-11-01970-f003:**
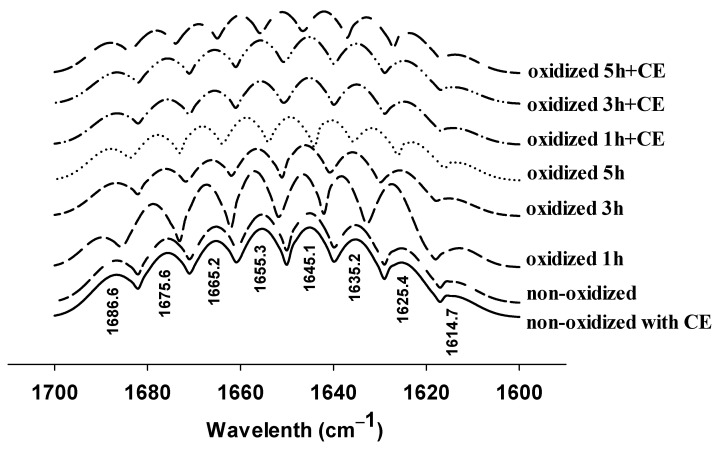
Influence of CE on Fourier transform infrared (FTIR) spectra of oxidized MPs during different oxidation times (0, 1, 3, and 5 h).

**Figure 4 foods-11-01970-f004:**
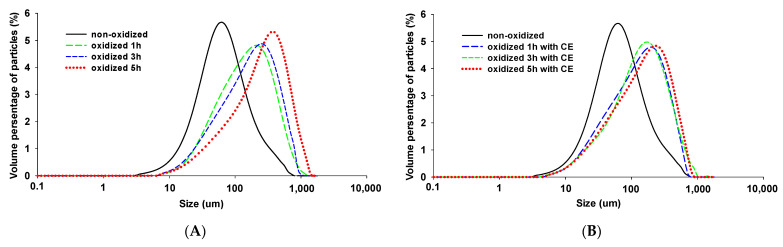
Particle size distribution of MPs at different oxidation times (0, 1, 3, and 5 h) (**A**) and CE under different oxidation times (**B**).

**Figure 5 foods-11-01970-f005:**
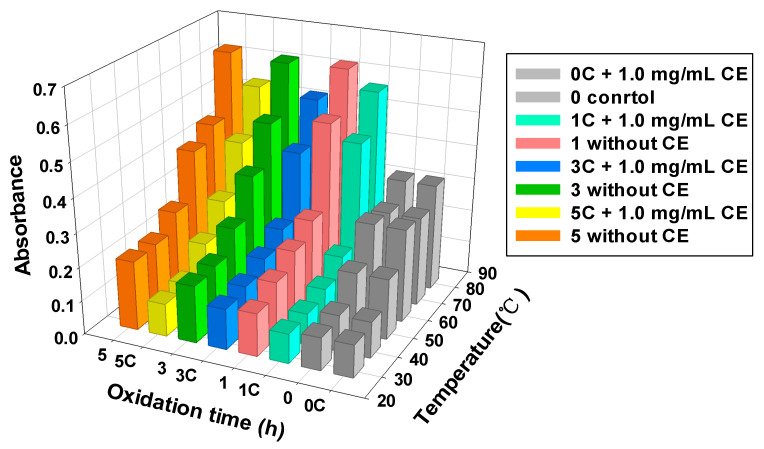
Effect of CE on MPs turbidity at different oxidation times. 0 and 0 C, not oxidized; 1 and 1 C, oxidized for 1 h; 3 and 3 C, oxidized for 3 h; 5 and 5 C, oxidized for 5 h.

**Figure 6 foods-11-01970-f006:**
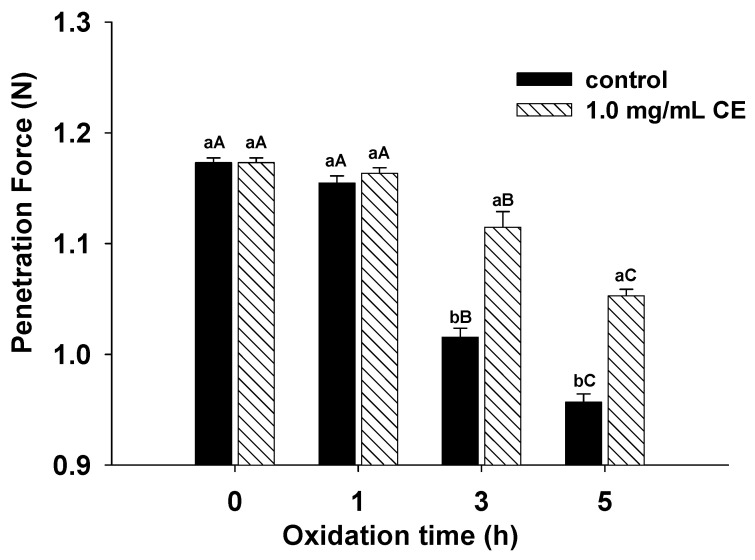
Effect of CE on MPs gel strength at different oxidation times (0, 1, 3, and 5 h). Different capital letters indicate the same treatment with significant differences at different oxidation times (*p* < 0.05); different lowercase letters indicate significant differences between different treatments under the same oxidation time (*p* < 0.05).

**Figure 7 foods-11-01970-f007:**
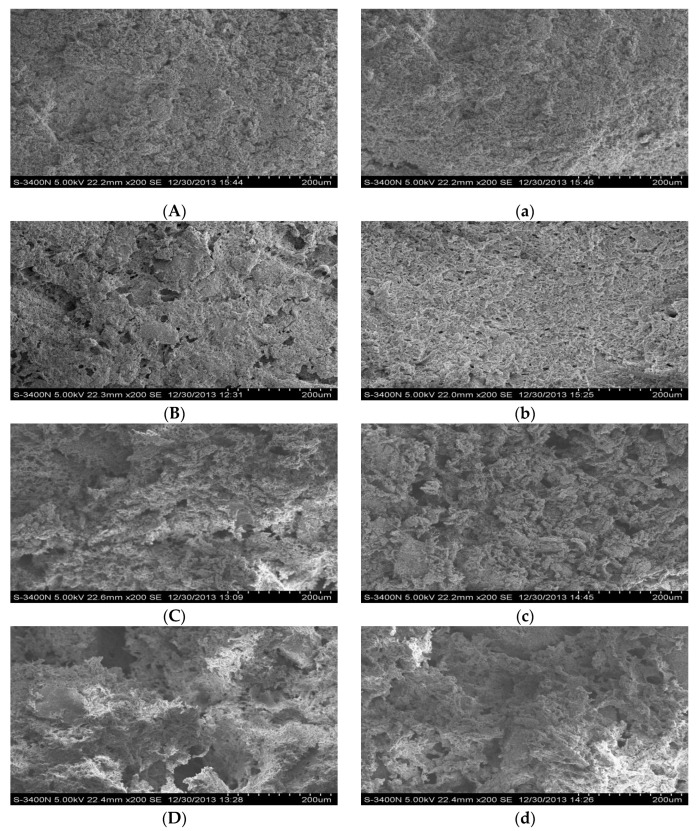
Electron micrographs of CE in the MPs gel at different oxidation times (Magnification 200×). (**A**) Non-oxidized; (**B**) oxidized for 1 h; (**C**): oxidized for 3 h; (**D**) oxidized for 5 h; (**a**) contains 1.0 mg/mL CE non-oxidized; (**b**) contains 1.0 mg/mL CE oxidized for 1 h; (**c**) contains 1.0 mg/mL CE oxidized for 3 h; (**d**) contains 1.0 mg/mL CE oxidized for 5 h.

**Table 1 foods-11-01970-t001:** Influence of CE on secondary structure content of oxidized MPs during different oxidation times (0, 1, 3, and 5 h).

Treatment	Oxidation Time (h)	α-Helix (%)	β-Sheet (%)	β-Turn (%)	Random Coil (%)	Aromatic Ring Vibration of Tyrosine Residues (%)
MPs samples without CE	0	17.21 ± 0.23 ^a^	46.44 ± 1.11 ^a^	14.81 ± 0.61 ^b^	17.41 ± 0.40 ^b^	4.11 ± 0.19 ^a^
	1	16.23 ± 0.25 ^ab^	47.91 ± 0.61 ^a^	14.82 ± 0.50 ^b^	17.64 ± 0.61 ^b^	3.42 ± 0.29 ^b^
	3	15.48 ± 0.21 ^bc^	47.31 ± 0.45 ^a^	14.94 ± 0.42 ^b^	17.63 ± 0.21 ^b^	4.44 ± 0.11 ^a^
	5	14.66 ± 0.39 ^c^	39.34 ± 0.33 ^b^	12.64 ± 0.50 ^c^	29.00 ± 0.42 ^a^	4.27 ± 0.07 ^a^
MPs samples with CE	0	17.19 ± 0.25 ^a^	46.31 ± 1.19 ^a^	14.77 ± 0.47 ^b^	17.40 ± 0.43 ^b^	4.08 ± 0.26 ^a^
	1	16.66 ± 0.35 ^a^	47.25 ± 0.82 ^a^	14.16 ± 0.38 ^bc^	17.53 ± 0.50 ^b^	4.39 ± 0.31 ^a^
	3	16.65 ± 0.35 ^a^	47.26 ± 0.94 ^a^	14.15 ± 0.37 ^bc^	17.52 ± 0.50 ^b^	4.40 ± 0.11 ^a^
	5	14.59 ± 0.55 ^c^	40.49 ± 1.06 ^b^	25.59 ± 0.99 ^a^	14.78 ± 0.29 ^c^	4.45 ± 0.22 ^a^

Notes: a–c means in the same column with different superscript letters differ significantly (*p* < 0.05).

## Data Availability

The data are contained within the article.

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
