# Peer review of "The Effectiveness of Clove Extract on Oxidization-Induced Changes of Structure and Gelation in Porcine Myofibrillar Protein"

_foods, 2022, doi:10.3390/foods11131970_

Round 1

Reviewer 1 Report

The article submitted by Ma and coworkers aimed to investigate the effectiveness of clove extract on the structural changes and gelation of porcine myofibrillar proteins induced by oxidation. This work is definitely of interest to the food chemistry and food oxidation fields, however, the following points should be considered:

1) I miss a clear statement from the authors about the rationale for choosing hydroxyl radicals to induce oxidation of MPs in their system. For example, it is well-known that other oxidants are also responsible for food oxidation or participate actively in the propagation of damage (e.g. peroxyl radicals). 

2) In introduction: In line 43 please remove "intramolecular" before carbonylation as carbonyls are generated in the sidechain of oxidizable residues and therefore it is not necessary to add that these are generated intramolecularly. Moreover, it is known that the oxidative cross-linking of proteins can trigger the formation of both inter-, as well as intra-molecular protein cross-links (https://doi.org/10.3390/molecules27010015), with both types of cross-links affecting protein structure. 

3) In materials and methods: 2.5 UV scanning spectra (Line 116): "Changes in the tertiary structure". This is not correct. The UV spectrum of proteins is not as sensitive to conformational changes as fluorescence, with Trp fluorescence being useful to determine structural changes in proteins (https://doi.org/10.1002/bmb.2002.494030030035). Moreover, some Tyr- and Trp-derived oxidation products also absorb light at 280 nm (https://doi.org/10.1016/j.freeradbiomed.2018.11.026; https://doi.org/10.1021/jf200277r)

This should be considered also in the results and discussion section as it is relevant for data interpretation. 

4) I would suggest the authors adding results about the formation of oxidation products in the sidechain of MPs. This would help to understand the pathways leading to MPs oxidation and aggregation, and the antioxidant effect/properties of the clove extract. For example, did the authors consider measuring the total protein carbonyls in their samples? or the total content of thiols? Running SDS-PAGE gels to determine the possible formation of intra-molecular cross-links on oxidation of MPs in the absence and presence of the clove extract? Or determine the possible formation of di-Tyrosine (this can be easily done by fluorescence spectroscopy, HPLC-FLD, or western blotting). These analyses would help explain the structural changes at the secondary structure level and the gelling properties of native versus oxidized samples, and would increase the relevance of the antioxidant mechanism allowing a better understanding of the chemical pathways. 

5) The discussion lacks profundity in the pathways that would be involved in the gelation, cross-linking and aggregation of oxidized MPs and the anti-oxidative pathways that would help decrease the oxidation of MPs. Are the aggregates stable or not? Are these generated as a consequence of covalent bonds? or non-covalent interactions? What is the mechanism associated with the antioxidant effect of the extract studied?

Other researchers have described that aggregation of MPs would occur via secondary reactions between Lys residues and carbonyl species. 

I believe this is a nice piece of work, but further experimental analyses are needed as the current version describes a phenomenon, but lacks the profundity of the pathways leading to this observation. 

Reviewer 2 Report

In this manuscript entitled " The Effectiveness of clove extract on oxididation-induced changes of structure and gelation in porcine myofibrillar protein", the authors were evaluating the inhibitory effect of clove extract on porcine myofibrillar protein oxidation. I think the data presented in this MS is very valuable. I have comments explained below. I hope that my comments are very useful for improving this research.

Comments
(1)    Title: oxididation -> oxidization
(2)    L38: Authors write that polyphenols are safe and non-toxic, but I believe this is incorrect. Ingredients found in foods can also be toxic depending on the amount ingested. Please correct the statement.
(3)    L163-165: Many data only show averages (e.g., Figure 1, 2, 3, 4, and 5). Therefore, this statement is incorrect. Please correct the statement.
(4)    L173: The study is limited by a lack of comparison group. Figures 1, 2, 3, 4, and Table 1 do not show non-oxidation with CE data. Therefore, when interpreting the results, without non-oxidation and CE data, the effects of CE alone cannot be determined and it is not clear whether CE truly has an antioxidant effect on myofibrillar protein.
(5)    L173: This experiment only evaluates CE. Therefore, I do not know if CE is highly effective. If there are previous reports, please add your discussion.
(6)    L173: There is no discussion of the oxidation inhibitor compound in CE. I believe the polyphenols in CE are active component, please discuss the oxidation inhibitor compounds in CE.
(7)    Figure 1B: As with Figure 1A, it should be colored.
(8)    L179-183: The authors write that oxidation changes amino acid composition, but this may be incorrect. I don't know what authors mean by this composition of amino acid. The phenomenon that occurs with oxidation in this study is probably a change in protein structure, not a change in amino acid composition. Please confirm this point.
(9)    L190-191: References needed.
(10)    L191-193: References needed.
(11)    L208-210: It is not that the tyrosine or tryptophan content has changed, but that the tyrosine or tryptophan has been altered by oxidation.
(12)    L225-226: The results of the statistical process should be shown in Figure 2. The figure should also indicate what statistical method was used.

Reviewer 3 Report

In the manuscript, the following points should be clarified:

In the material and method section, the experimental design of the research should be explained. 

Information should be given about the pH of the raw material. How many different animal muscles were used in the study, or was just the same muscle used from one ?

Statistical analysis: Randomized complete block design or completely randomized design?

Looking at Table 1, there is no statistical difference between the groups with and without CE for alpha helix and beta sheet. In this case, is it possible to say that CE is very effective?

Reviewer 4 Report

Line 89: Please specify the size of 24-mesh sieve as micron. 

Line 103: Please check the sentence. There is a grammatical problem.

Line 127-134: It is not clear how to prepare samples for FT-IR analysis. As to be other analysis, the samples were in liquid form. Did authors measure the FT-IR in liquid form, or dry form?

Line 142: Instead of "measured" please write "held".

Line 305: Please check the sentence. I think the brackets should be removed.

In the results section of Particle size and Turbidity, the authors explained the increasing turbidity with reducing thermal stability of MPs. For making this kind of explanation, the thermal properties of MPs with or without CE should be investigated. The increasing temperature may indicate the increasing solubility and hence decreasing turbidity, not increasing thermal stability. It is better to explanain thermal stability with thermal analysis such as DSC and/or TGA. 

Line 346-364: The content of α-helix and  β-sheet have significant effect on the gel strength properties. Please collaborate the results and make some additional explanations. 

Round 2

Reviewer 1 Report

The authors have replied to all my questions and have substantially improved the article. 

Reviewer 2 Report

I am satisfied with the revisions that have been made by the authors.